# DRFM-Based Repeater Jamming Reconstruction and Cancellation Method with Accurate Edge Detection

Bowen Han [1] , Xiaodong Qu [1,*], Xiaopeng Yang [1,2], Wolin Li [1] and Zhengyan Zhang [1]

1   School of Information and Electronics, Beijing Institute of Technology, Beijing 100081, China
2   Yangtze Delta Region Academy of Beijing Institute of Technology, Jiaxing 314019, China
*   Correspondence: xdqu@bit.edu.cn; Tel.: +86-010-6891-8377

**Abstract:** Digital radio frequency memory (DRFM) based repeater jamming can create false targets, which can lead to a loss of situational awareness, misidentification of targets, and decreased overall performance of the radar system. Traditional jamming suppression methods do not give due importance to the accurate estimation of the jamming edge, resulting in jamming residual and poor anti-jamming performance. To tackle this issue, this paper explores the reason and impact of inaccurate jamming edge estimation and proposes a DRFM-based repeater jamming reconstruction and cancellation method with accurate edge detection. In the proposed method, firstly, multiple jamming parameters are obtained by computing the short-time fractional Fourier transformation (*STFRFT*) spectrogram of the received signal. To avoid jamming residue, the proposed method estimates the accurate jamming edges by the joint use of the difference of box (DOB) filters and time domain deconvolution (TDDC) curves. Numerical simulations and experiments are conducted to evaluate the algorithm's effectiveness in countering smeared spectrum (SMSP) and interrupted sampling repeater jamming (ISRJ). The results demonstrate its superior jamming reconstruction and suppression performance than other methods.

**Keywords:** digital radio frequency modulation; repeater jamming; jamming reconstruction and cancellation; accurate edge detection





## 1. Introduction

With the development of digital radio frequency memory (DRFM) technology and devices, lots of DRFM-based repeater jamming types are proposed and widely used in electronic countermeasure scenarios. During them, SMSP jamming is first proposed by Sparrow in 2006 [1], and the concept of ISRJ is first proposed by Wang in 2007 [2]. These two types of jamming, referred to as DRFM-based repeater jamming in this article show excellent performance in countering linear frequency modulation (LFM) radar. Jammers can obtain pulse compression (PC) processing gain and exceed other incoherent jamming types in output power. The intermittent sampling process allows the jammer to transmit the jamming without receiving a complete signal and can enhance the immediacy of jamming [2,3]. The jamming creates a great number of false targets in the results after pulse compression according to the Fresnel ripple effect, and may seriously affect the target detection performance of the radar. Due to these reasons, it is urgent to study the countermeasures of DRFM-based repeater jamming.

In recent years, various methods for combating DRFM-based repeater jamming have been published, which can be categorized into two main aspects: signal filtering and jamming reconstruction and cancellation. The signal filtering methods are aided by time-frequency transformation tools to generate filters, such as short-time Fourier transform (STFT) [4], fractional Fourier transform (FRFT) [5], and empirical mode decomposition (EMD) [6]. These methods typically utilize the discontinuity characteristic of the jamming signal to separate the jamming and target signal. However, the inherent error of time-frequency transformation tools leads to inaccurate filter generation, which in turn results

in jamming residual or signal loss after filtering. A more recent method [7] combine neural networks and STFT to train filters, but this method suffers from a huge amount of computation and limited model generalization performance. In order to combat jamming that cannot be completely filtered out in the time-frequency domain, various methods are proposed using multi-domain information to separate jamming and target signal, such as methods using information from space [8], polarization [9] and waveform domain [10,11]. However, these methods require the specialized design of the radar system or coordination with the radar transmitter, which increases the complexity of the radar system, introduces more errors, and limits the application scenarios.

The jamming reconstruction and cancellation method of DRFM-based repeater jamming mainly contains two steps. The first step is jamming parameter estimation, which comprehensively utilizes the parameters of the radar transmission signal and the jamming information extracted from the echo signal. The following steps are jamming signal reconstruction and cancellation using the estimated jamming parameters. Jamming parameters estimation usually uses time-frequency analysis tools including STFT [12], matched signal transform (MST) [13], FRFT [14–16] and other signal separation methods including blind source separation [17,18], compressed sensing [19] and deep-learning [20]. However, these methods focus on estimating the amplitude, frequency modulation slope and time width parameters of the jamming, while ignoring the accurate estimation of the interference edge. This leads to jamming residual after cancellation, which in turn affects the radar performance.

In this paper, a novel jamming suppression method for DRFM-based repeater jamming based on jamming reconstruction and cancellation is proposed. First, the optimal FRFT order of the jamming is obtained by two-step searching. The center frequency of each jamming slice and the frequency modulation rate of the jamming are also calculated. Second, the fractional time-frequency plane is obtained by *STFRFT* under the optimal order, and the number and position of slices are extracted. Then, the difference of box (DOB) filter and time domain deconvolution (TDDC) of the received signal is performed sequentially to obtain the accurate edge of the jamming slice. Finally, the jamming signal is reconstructed and canceled from the received signal to obtain the anti-jamming output.

The rest of the paper is organized as follows. In Section 2, the transmitting and jamming signal model used in this paper is introduced. In Section 3, the disadvantage of inaccurate jamming edge estimation in jamming suppression is analyzed and the proposed jamming suppression method is introduced in detail. Numerical simulation and experiment results are provided in Section 4 to validate the effectiveness of the method, and the conclusion is drawn in Section 5.

## 2. Jamming Signal Model

In this section, the jamming signal model of ISRJ and SMSP jamming based on the LFM radar transmitting signal has been given and analyzed.

The jamming principle of ISRJ is shown in Figure 1. The jammer first intercepts a slice of the radar transmitting signal and forwards the slice; then, the above process is repeated until the back edge of the signal is detected.

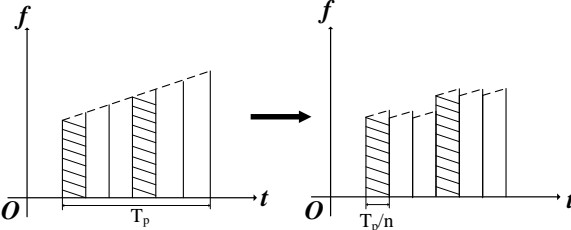

**Figure 1.** The principle of ISRJ.

The transmitted LFM signal is given by

$$S_t(t) = A_t \text{rect}\left(\frac{t}{T_p}\right) e^{[j2\pi(f_0 t + \frac{1}{2}kt^2)]} \tag{1}$$

where $A_t$ denotes the amplitude of the transmitted signal, $k = \frac{B}{T_p}$ denotes the frequency modulation rate of the signal, $B$ denotes the bandwidth, $T_p$ denotes the pulse width, $f_0$ denotes the carrier frequency, and $\text{rect}\left(\frac{t}{T_p}\right)$ denotes a rectangular window function with the duration of $T_p$.

For the LFM signal, the slice intercepted by the jammer can be modeled as

$$J_c(t) = A_J \sum_{n=0}^{N_c-1} \text{rect}\left(\frac{t - \tau - T_n}{T_J}\right) \cdot e^{[j2\pi(f_0(t-\tau) + \frac{k}{2}(t-\tau)^2)]} \tag{2}$$

where $A_j$ denotes the amplitude of the jamming signal, $\tau$ is the delay of jamming, $T_n$ is the interception start time of the slice, and $n = 0, 2 \cdots, N_c - 1$ are the slice numbers.

The intercepted slices are then forwarded $M$ times; the complete form of the jamming signal can be expressed as

$$J(t) = A_J \sum_{m=1}^{M} \sum_{n=0}^{N_c-1} \text{rect}\left(\frac{t - \tau - T_n - mT_J}{T_J}\right) \cdot e^{[j2\pi(f_0(t-\tau-mT_J) + \frac{1}{2}k(t-\tau-mT_J)^2)]} \tag{3}$$

After pulse compression, the jamming signal appears as multiple false target groups in the range direction; each false target group consists of a main false target and several symmetrically distributed secondary false targets. By changing the slice width and forwarding times, the jamming can achieve effects of both deception and suppression.

The principle of SMSP jamming is shown in Figure 2. As a DRFM-based repeater jamming, the jammer also intercepts and forwards certain jamming slices. However, a typical SMSP jamming slice is intercepted by sampling the original LFM radar signal with n times sampling interval, and the forwarding process is repeated n times. Thus, the jamming pulse modulation rate is n times the original signal.

Consider the SMSP jamming containing $n$ slices. The SMSP jamming can be expressed as

$$J_{SMSP}(t) = A_J \sum_{i=0}^{n-1} \text{rect}\left[\frac{n\left(t - \frac{iT_p}{n}\right)}{T_p}\right] e^{j2\pi[f_0(t-\frac{iT_p}{n}) + \frac{k_{SMSP}}{2}(t-\frac{iT_p}{n})^2]} \tag{4}$$

where $k_{SMSP} = nk$ denotes the frequency modulation rate of the jamming slice.

In a practical situation, the jammer may not intercept the full pulse of the radar transmit signal. In addition, the jammer may not continuously forward the slices. According to the two cases above, a more general expression of SMSP jamming is given as

$$J_{SMSP}(t) = A_J \sum_{i=0}^{n-1} \text{rect}\left[\frac{n\left(t - \frac{iT_c}{n} - T_i\right)}{T_c}\right] \cdot e^{j2\pi[f_0(t-\frac{iT_c}{n}-T_i) + \frac{k_{SMSP}}{2}(t-\frac{iT_c}{n}-T)^2]} \tag{5}$$

where $T_c$ is the jamming intercept pulse width, $T_i$ is the extra time delay of the $i$-th jamming slice, and $k_{SMSP} = \frac{nB}{T_c}$.

After pulse compression, each slice of the SMSP jamming will form a group of equally spaced false targets before and after the time delay, and the amplitude of each false target is close. Although SMSP jamming cannot obtain complete pulse compression gain of the victim radar, each false target still has a spectral density distribution similar to the real target, making the jamming difficult to be distinguished when the jamming energy is greater than the target energy.

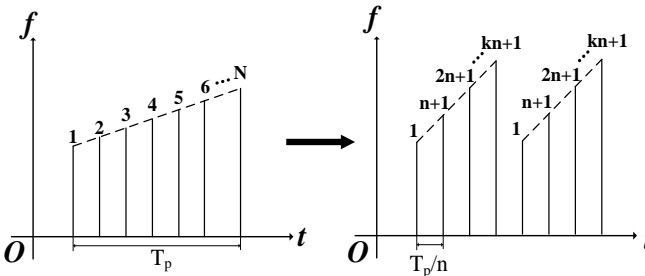

**Figure 2.** The principle of SMSP jamming.

## 3. Proposed Jamming Cancellation Method

In this section, a jamming cancellation method based on jamming parameter estimation and jamming reconstruction is proposed in detail. The performance of jamming reconstruction and cancellation strategy heavily depends on the sufficiency and accuracy of jamming parameters estimation. However, it is difficult to accurately estimate the jamming parameters. In addition, SMSP jamming has a different frequency modulation slope from the target signal, which brings more challenges in parameter estimation. Based on the principle of SMSP jamming generation, the reconstruction of jamming relies on the accurate estimation of the modulation rate, jamming slice numbers, pulse edge, pulse width, and center frequency.

In the proposed method, the frequency modulation rate is first obtained by searching optimal fractional order in FRFT. Then the center frequency and pulse width are roughly estimated using edge detection in the *STFRFT* domain. Furthermore, the DOB filtering is performed to reveal the accurate position of jamming slice edges. Finally, the edge detection result is refined by time domain deconvolution. The complete process of the proposed algorithm is shown in Figure 3.

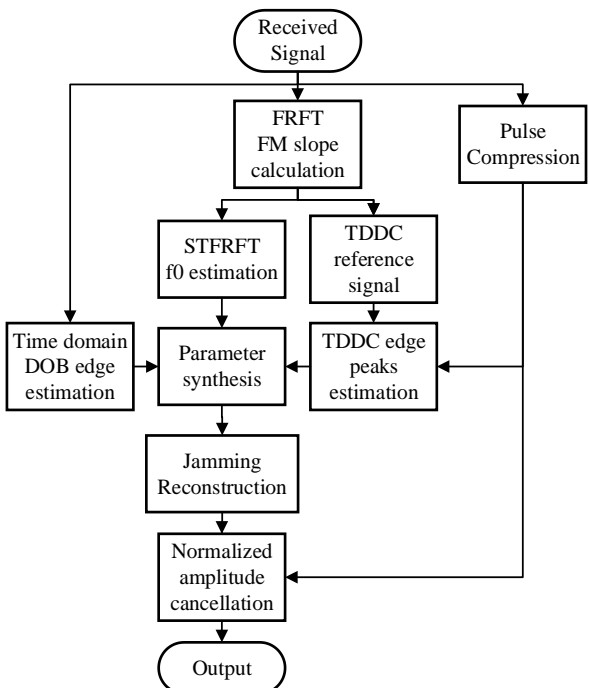

**Figure 3.** Block diagram of the proposed method.

The proposed method is raised under the condition that the power of jamming is higher than that of target echo after PC and has a positive interference-to-noise ratio (INR), which is common in the working environment of an anti-jamming environment. If the input INR is low, denoising methods need to be applied first to ensure jamming detection.

The details of the jamming reconstruction and cancellation method are described in the following section.

### 3.1. Jamming Parameter Estimation

The FRFT of a signal can be regarded as its Wigner–Ville distribution with rotated by angle $\alpha$ which is corresponding to the order of FRFT $p$. Therefore, the FRFT of an LFM signal becomes an impulse function in fractional frequency domain once the FRFT order is aligned with the frequency modulation slope, which is convenient for signal detection and parameter extraction. If $\alpha = -\text{arccot} k_{SMSP}$, multiple high-power peaks associated with SMSP jamming slices can be observed in FRFT. Basically, the FRFT of the received signal $S_0(t)$ is

$$S_\alpha(u) = \int_{-T/2}^{T/2} K_p(t, u) S_0(t) dt = \sqrt{\frac{1 - j \cot \alpha}{2\pi}} \int_{-T/2}^{T/2} e^{j[(u^2 + t^2) \cot \alpha / 2 - ut \csc \alpha]} S_0(t) dt \quad (6)$$

Note, that the jamming power is typically stronger than the target echo, we can obtain the frequency modulation rate of the jamming by searching the order of FRFT that corresponds to the highest peaks

$$p^* = \text{argmax}\{|F^p[s_t(t) + J_{SMSP}(t)]|\} \quad s.t \; p \in [-2, 2] \quad (7)$$

$$\widetilde{k} = -\cot(\frac{\pi}{2} p^*) \quad (8)$$

Thereafter, the jamming slice parameters can be obtained by *STFRFT* using the optimum order $p^*$.

A fine search step is required in optimum order search to achieve enough accuracy, but it significantly increases the computational burden as FRFT needs to be conducted in each cycle. In this work, we adopt a two-step searching strategy to reduce computational complexity. As the SMSP jamming modulation slope is always greater than the known target signal, the searching can be applied within $[-2\text{arccot}k/\pi, 2]$ only. First, we coarsely sweep the order number $p$ to roughly locate the peaks with a larger search grid $\varepsilon_{coarse}$, then finely tuned $p$ to achieve more accurate estimation within adjacent grids.

Once the optimum order of FRFT is found, it can be used in *STFRFT* to distinguish different SMSP pulses and estimate the center frequency of each jamming pulse. The center frequency is used in the reconstruction process to determine the start frequency of each jamming slice since the jamming slice can be intercepted from any part of the transmitting signal. Similar to the extension of the Fourier transform to the fractional Fourier transform, *STFRFT* is also the extension of the short-time Fourier transform, which can be considered as a windowed fractional Fourier transform [15].

The short-time fractional Fourier transform of signal $x(t)$ can be defined as

$$STFRFT_{(x,p)}(t, u) = \int_{-\infty}^{+\infty} x(\tau) g(t - \tau) K_p(t, u) d\tau \quad (9)$$

where $g(t)$ is the window function.

From the *STFRFT* spectrogram, a jamming slice number can be obtained. Figure 4 shows the *STFRFT* spectrogram for four slice jamming, from which four jamming pulses can be seen. The center frequency parameter of each jamming slice can also be extracted from the *STFRFT* spectrogram. First, the fractional frequency $u_0$ of each slice is extracted by centroid peak detection from the fractional frequency accumulating results of the spectrogram. Then, $u_0$ can be converted into the center frequency $f_0$ by

$$f_0 = \frac{u_0}{\sin \alpha} \quad (10)$$

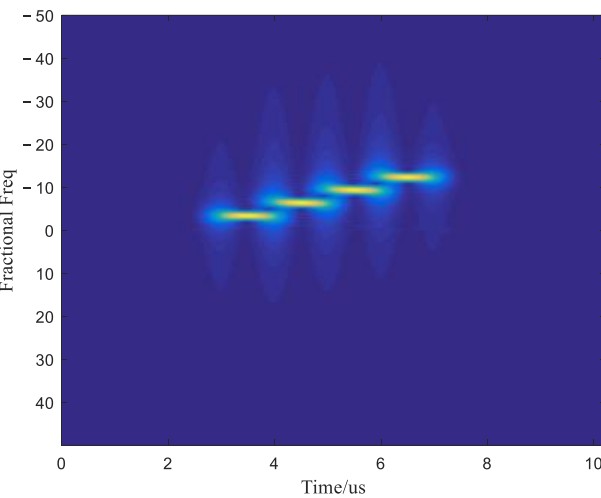

**Figure 4.** *STFRFT* spectrogram of a 4-slices jamming.

### 3.2. Jamming Edge Estimation

The edge extension phenomenon in STFT and *STFRFT* spectrograms makes it hard to accurately measure the accurate pulse width and edge positions [21]. That is because the essence of STFT and *STFRFT* is to take a fixed width time domain window at each time point, and then FFT or FRFT is applied to the signal in each window. Without losing generality, take the *STFRFT* spectrogram as an example. When *STFRFT* is calculated at the slice edge, only part of the signal enters the time window. Under this circumstance, the *STFRFT* is converted into a partial signal FRFT problem, and the results are shown in Figure 5. It can be seen from the figure that the peak formed by the jamming edge FRFT result (in the lower part of the figure) is lower than that of the jamming center, and the extension occurs. In addition, as long as the time window contains part of the jamming slice, the *STFRFT* result exists in the corresponding time unit. Therefore, the time range in which the jamming can be detected is also wider than the actual signal edge, which leads to the edge extension in the time domain. In theory, the edge extension can be changed by adjusting the length of the *STFRFT* window. In practice, the narrower the time window of *STFRFT* is, the worse the resolution of the frequency axis is. This will lead to the failure of the center frequency estimation process.

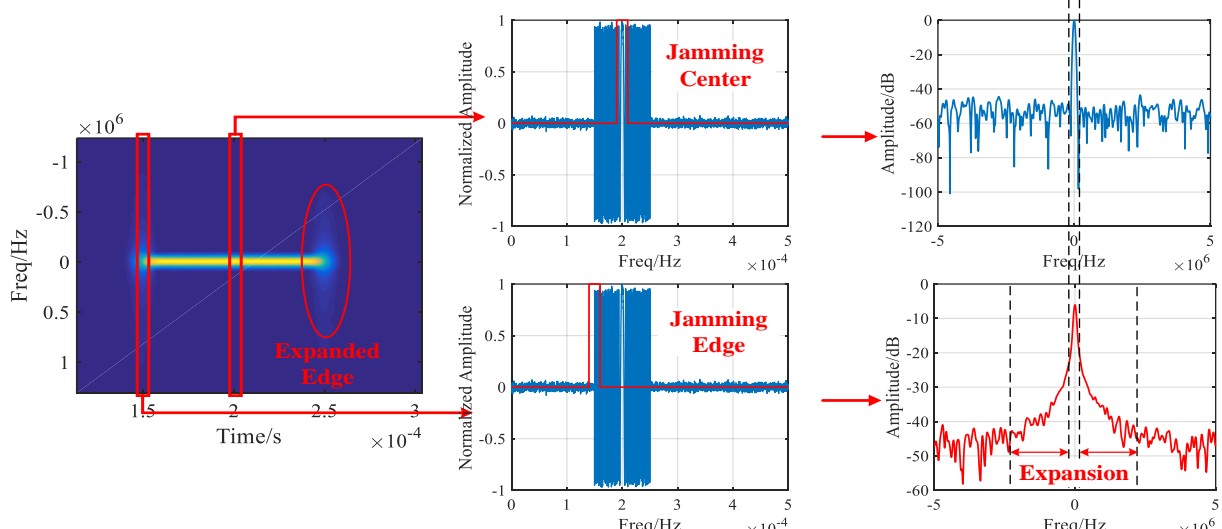

**Figure 5.** Edge expansion phenomenon in *STFRFT* spectrogram.

The jamming edge estimation error will produce a jamming residual after reconstruction and cancellation which will cause a serious negative impact on the anti-jamming performance. The result of a typical jamming cancellation result after pulse compression with estimation error in the front edge and back edge are, respectively, shown in Figure 6. In order to more clearly show the performance degradation caused by the edge estimation error, in this example, the reference signal used in pulse compression is with the same frequency modulation slope as the jamming. Therefore, the pulse compression result of the interference slice is a single peak. For the PC envelope cancellation method used in this paper, the estimation of the front edge is not accurate, and the peak positions of reconstructed jamming do not coincide with that of the real jamming. As a result, jamming residues occur heavily after cancellation, as shown in Figure 6a. However, when the back edge estimation is not accurate, there is only less jamming residue after cancellation, as shown in Figure 6b.

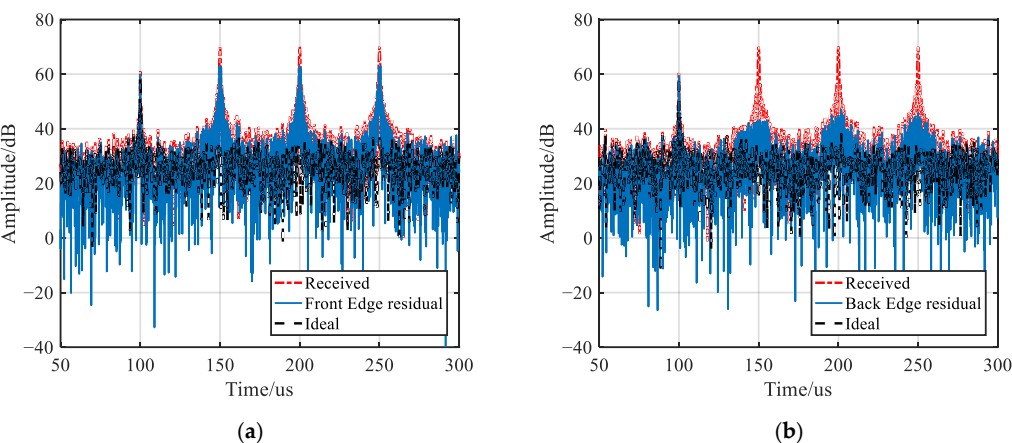

(a)　　　　　　　　　　　　　　　　　　　　　(b)

**Figure 6.** Influence of edge estimation error of: (**a**) Front edge; (**b**) Back edge.

Different pulse edge detection methods have been studied regarding various types of radar signal. In this paper, the Difference of Box (DOB) filter [22] with *STFRFT* for pulse width detection is adopted since DOB is one of the most robust edge detection filters under Canny's criteria and it can take the advantage of the temporal time-frequency analysis capability. Figure 7 shows the edge detection results of DOB with different filter lengths in different jamming cases. The signal to be detected is a four slice jamming. Note, that if the jamming pulses are separated in the time domain (jamming pulses forward discontinuously), the edge detection filter can clearly identify the edges, as shown in Figure 7a; but once the pulses are close to the edge, as shown in Figure 7b, the filter output can only approximately indicate the existence of the edges.

As can be seen above, the effectiveness of DOB is limited in handling continuous jamming slices. Therefore, this paper introduces the time domain deconvolution (TDDC) method as a supplementary approach to DOB edge estimation [23]. The proposed TDDC method is essentially applying Weiner filtering assuming the point spread function (PSF) is the original jamming signal. The deconvoluted PC results are a set of impulses that indicate the beginning and end of each jamming slice.

A virtual jamming signal is constructed by using the rough estimation of jamming parameters from *STFRFT* and DOB to form the PSF. The virtual jamming signal is expressed as

$$\widetilde{S}_t(t) = \text{rect}\left(\frac{t}{\widetilde{T}_p}\right) e^{j2\pi(f_0 t + \frac{1}{2}\widetilde{k}t^2)} \tag{11}$$

where $\widetilde{k}$ is calculated from (9), $f_0$ is calculated from (11), and $\widetilde{T}_p$ is the rough estimation of jamming pulse $T_p$ by DOB filtering.

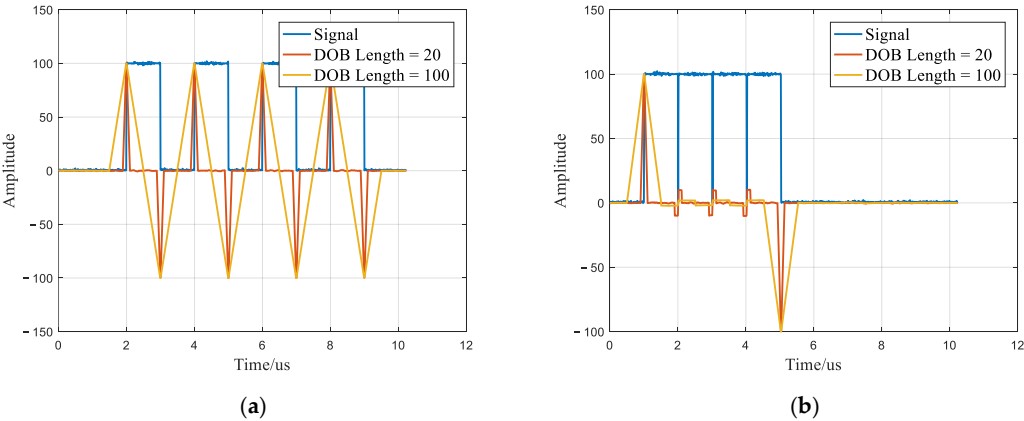

**Figure 7.** DOB edge detection curves of jamming with (**a**) Jamming pulses forward discontinuously; (**b**) Jamming pulses forward continuously.

The PSF of the virtual jamming signal is used to deconvolute the jamming PC result. The deconvolution signal can be expressed as

$$S_{dc} = \text{ifft}\left[\frac{\text{fft}(J) \cdot \text{fft}^*(\widetilde{S}_t)}{\text{fft}(\widetilde{S}_t) \cdot \text{fft}^*(\widetilde{S}_t)}\right] = \text{ifft}\left[\frac{\text{fft}(J)}{\text{fft}(\widetilde{S}_t)}\right] = \text{ifft}\left[\frac{F_J}{F_{\widetilde{S}_t}}\right] \tag{12}$$

where $F_{\widetilde{S}_t}$ and $F_J$ are the spectrum of the virtual transmitted signal and jamming signal. The spectrum of the deconvolution signal $\text{fft}(S_{dc}) = \frac{F_J}{F_{\widetilde{S}_t}}$ can be abbreviated as

$$\frac{F_J}{F_{\widetilde{S}_t}} = \frac{2\theta}{jE_t\pi}e^{j\pi\alpha/2}[\cos(\pi\beta_1 f) + \cos(\pi\beta_2 f)] - \frac{4\theta}{j\pi} \tag{13}$$

where $E_t = E\left(\frac{k_{SMSP}T_p + 4f}{2\sqrt{2k_{SMSP}}}\right)^2 - E\left(\frac{k_{SMSP}T_p - 4f}{2\sqrt{2k_{SMSP}}}\right)^2$, in which $E(X)$ denotes complex Fresnel integral. $\alpha = k_{SMSP}(T_p^2 + T_J^2)/2$, $\beta_1 = T_p + T_J$, $\beta_2 = T_p - T_J$ and $\theta = \arctan(T_J/T_p)$.

Since the IFFT of the cosine function is an impulse function, the deconvolution result of the SMSP jamming slice forms a series of impulses with different amplitudes, which are extended periodically at

$$t = \pm n\beta_1, \ t = \pm n\beta_2, \ n = 1, \ 2, \ \cdots \tag{14}$$

Among the impulses, two impulses can always be formed at the rising and falling edge of the jamming slice. Figure 8 shows the TDDC curve of four slice jamming. It can be observed that multiple periodic peaks are exhibited in the curve, with each jamming slice's front and back edges corresponding to one of the peaks on the curve.

In comparison to the edge detection results of DOB filters, the energy of TDDC impulses is more concentrated, which results in higher estimation accuracy. However, the peak distribution of the curve generated by TDDC is periodic and unable to determine the location of the jamming edge on its own. Thus, we combine DOB with TDDC to achieve an optimal estimation of the jamming edge. Firstly, DOB is used to obtain the edge position estimation results of all jamming slices. Then, the peak closest to the DOB estimation result is extracted from the TDDC curve to provide an accurate estimation of the jamming edge.

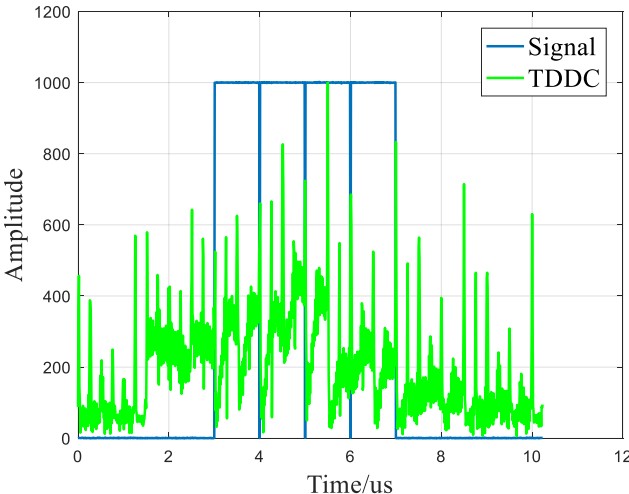

**Figure 8.** The TDDC curve of four slice jamming.

### 3.3. Jamming Reconstruction and Cancellation

The PC amplitude response of the jamming is determined by the time width of jamming slices, the number of slices and the center frequency of slices. Therefore, the jamming signal can be reconstructed after the three parameters are estimated and then canceled from the received signals to suppress the jamming. In the proposed method, the amplitude envelope normalized cancellation method after PC is applied, which can avoid the influence of amplitude error and phase error in the estimation process.

The received signal is firstly pulse compressed, and its amplitude envelope is normalized as

$$X_0 = \frac{\text{abs}(\text{ifft}[\text{fft}(S_0) \cdot \text{fft}^*(S_t)])}{\max(\text{abs}(\text{ifft}[\text{fft}(S_0) \cdot \text{fft}^*(S_t)]))} \tag{15}$$

Noticing that different jamming slice has a different center frequency, the jamming signal is reconstructed as

$$S_{rec}(t) = \sum_{j=1}^{N_J} \text{rect}\left(\frac{t}{T_j}\right) e^{j2\pi(f_j(t-t_j) + \frac{1}{2}\tilde{k}(t-t_j)^2)} \tag{16}$$

where $N_J$ is the estimated slice number from *STFRFT*, $T_j$, $t_j$ are the estimated time width and front edge time of the *j*-th jamming slice, and $f_j$ is the estimated center frequency of the *j*-th jamming slice.

For applying jamming cancellation, the reconstructed jamming signal is also pulsed compressed, and the amplitude envelope of the range profile of pulse compression is normalized as

$$X_{rec} = \frac{|\text{ifft}[\text{fft}(S_{rec}) \cdot \text{fft}^*(S_t)]|}{\max(|\text{ifft}[\text{fft}(S_{rec}) \cdot \text{fft}^*(S_t)]|)} \tag{17}$$

Finally, the reconstructed jamming PC amplitude envelope is canceled from the received signal amplitude envelope of the one-dimensional range profile. The output amplitude envelope is expressed as

$$X_{out} = X_0 - X_{rec} \tag{18}$$

It can be analyzed that the phase error of the estimate jamming does not affect the cancellation result, as only the PC amplitude is considered in the target detection. Furthermore, because the proposed method is applied to the received signal within each PRT in the fast time domain, the Doppler effect can be neglected.

## 4. Numerical Simulation and Experiment Analysis

In this section, the performances of the proposed method are investigated by numerical simulations and experiments in terms of jamming reconstruction accuracy and jamming suppression effect. The results are shown as follows.

### 4.1. Numerical Simulations

In the simulation, an S-band LFM radar is considered. Assuming that the receiving signal of radar contains one target signal and one repeater jamming signal. The jamming signal is overlapped with the target signal but is incoherent, which means that the false target peak formed by the repeater jamming does not coincide with the target peak. The detailed simulation parameters are given in Table 1.

**Table 1.** Simulation parameters.

| Parameters | Value |
|:---:|:---:|
| Carrier frequency | 3 GHz |
| Bandwidth | 5 MHz |
| Pulse width | 200 μs |
| Sampling rate | 10 MHz |
| Target distance | 3000 m |
| ISRJ slice width | 50 μs |
| ISRJ forwarding times | 3 |
| SMSP sampling rate | 20 MHz |
| SMSP forwarding times | 4 |
| SNR | 0 dB |
| INR | 30 dB |

The edge detection method based on the DOB filter and TDDC is first evaluated. The modulation rate of the jamming is obtained by the two-step searching of the optimal rotation angle of FRFT. Then the DOB filter and TDDC are applied on the *STFRFT* in the optimal order. The TDDC result compared with the DOB result is shown in Figure 9. It can be seen that TDDC indeed improves the accuracy of edge detection. TDDC peaks out of the jamming range are not concerned with the edge detection since they are not related to the DOB peaks.

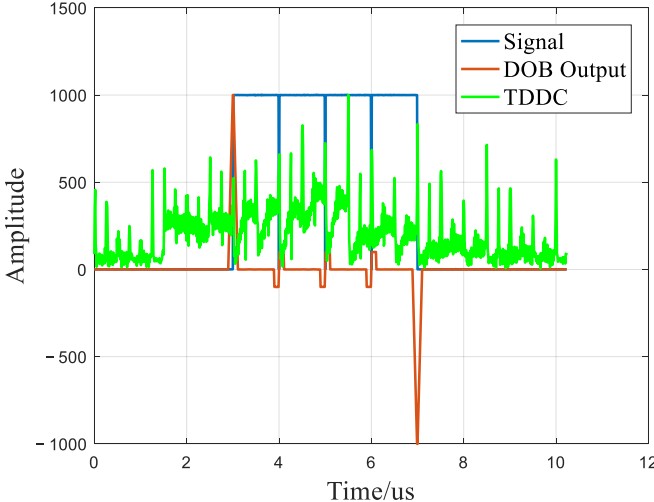

**Figure 9.** TDDC result compared with DOB result.

In order to measure the accuracy of the edge detection method, the starting time and the jamming pulse width are selected as the measurement parameters, and the MSE of the estimation results of these two parameters are calculated and compared with other

methods. The MSE of the pulse width and the starting time as a function of INR is shown in Figure 10. It can be seen that the MSE of the proposed method can reach under −20 dB, while the MSE of comparison methods remains at a high level and varies slightly when INR improves. It means the proposed method can obtain more accurate estimation results of edge parameters, resulting in better jamming suppression performance.

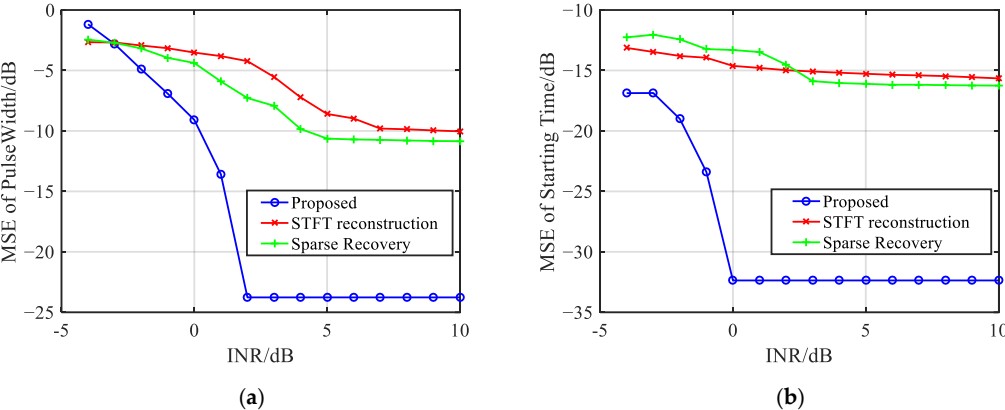

**Figure 10.** Comparison of the MSE calculated from: (**a**) Pulse width estimation; (**b**) Starting time estimation.

By knowing the slice position, slice pulse width, center frequency and frequency modulation rate, the jamming signal can be reconstructed. Figure 11 compares the mean square error (MSE) between the estimated value and real jamming of the proposed method with the FRFT filtering method in [5], the STFT jamming reconstruction and cancellation method in [12] and the sparse recovery method in [19]. It shows that no matter ISRJ or SMSP jamming, the MSE of the proposed method is better than that of other methods.

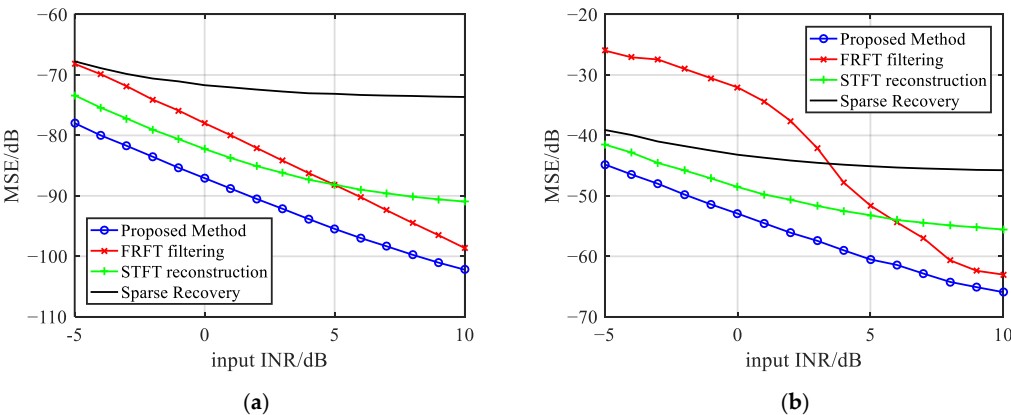

**Figure 11.** Comparison of jamming reconstruction MSE of the proposed method with other methods. (**a**) ISRJ reconstruction; (**b**) SMSP jamming reconstruction.

Once the jamming is reconstructed, it can be canceled from the received signal. The jamming suppression performance of the proposed method is evaluated by signal-interference-to-noise ratio (SINR) improvement. The SINR calculation is proceeded by using the pulse compression results before and after jamming suppression. The calculation method of SINR is defined in (19), and the SINR improvement is defined as (20), where $P_s$ refers to the output signal power, $P_J$ refers to the jamming power and $P_N$ refers to the noise power; $SINR_{after}$ represents the SINR after jamming suppression and $SINR_{received}$ represents the SINR of the received signal.

$$SINR = 10 \times \log 10(\frac{P_s}{P_J + P_N}) \tag{19}$$

$$I_{SINR} = SINR_{after} - SINR_{received} \tag{20}$$

Figure 12 plots the one-dimensional range profile of the output signals after applying the proposed method and other methods. With the proposed method, the output only contains the target echo, while the output of other methods has jamming residues caused by the inaccurate estimation of jamming. It can also be seen in the figure that the FRFT filtering method cannot form an accurate target peak, which is caused by the restriction of the grid error in the fractional order searching process. The calculated MSE of the target in the proposed method is −66.2 dB, while the MSE of other methods can only reach a minimum of −63.5 dB. The SINR improvement of the proposed method can reach more than 16 dB, while other methods are less than 15 dB. The results indicate that the proposed algorithm yields a higher performance improvement than other methods.

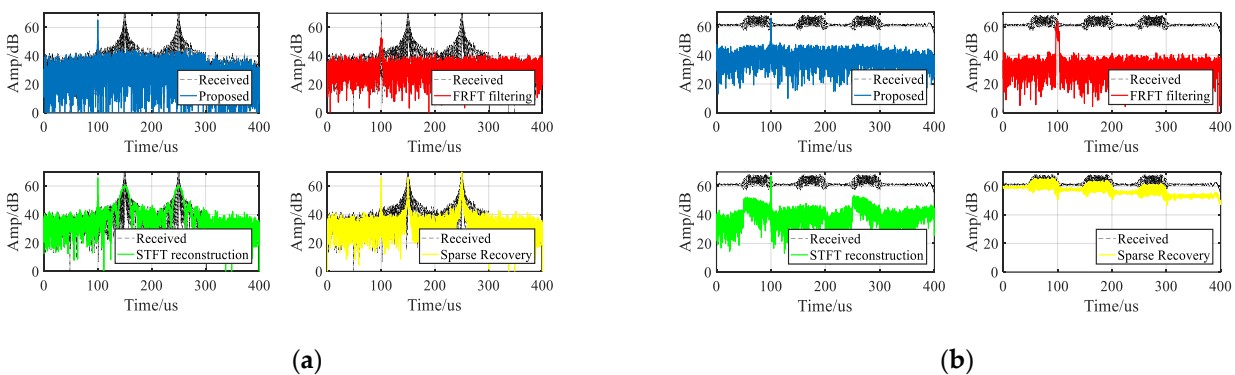

**Figure 12.** Comparison of the pulse compression range profile of the proposed method, FRFT filtering, STFT reconstruction and Sparse Recovery. (**a**) ISRJ; (**b**) SMSP jamming.

To intuitively show the jamming suppression performance, the comparison of the SINR improvement changing with input INR of the proposed method with FRFT filtering, STFT reconstruction and sparse recovery methods are shown in Figure 13. It can be seen from the figure that the SINR improvement of the proposed method is better than other methods. The computational complexity of the proposed method and the other methods has also been analyzed. The computational complexity of the proposed method is at the same order of magnitude of STFT reconstruction and sparse recovery, but it achieves better jamming edge estimation accuracy and the jamming suppression performance is better.

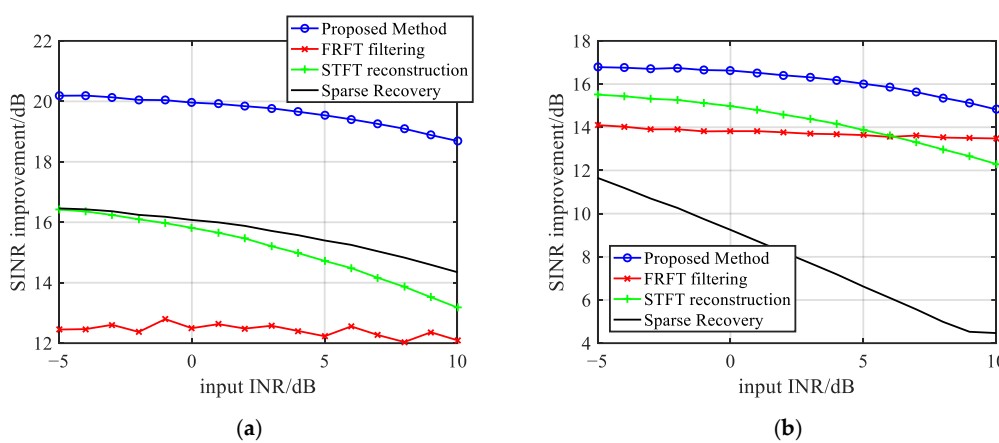

**Figure 13.** Comparison of SINR improvement of the proposed method with FRFT filtering, STFT reconstruction and Sparse Recovery. (**a**) ISRJ; (**b**) SMSP jamming.

The computational complexity of the proposed method and other methods has been analyzed and listed in Table 2. The computational complexity of the proposed method is similar to that of STFT reconstruction and sparse recovery. Though the proposed method is more complex than FRFT filtering, it achieves better jamming suppression performance.

**Table 2.** Computational complexity comparison of various methods.

| Method | SINR Improvement (dB) |
|---|---|
| proposed method | $O((M + P + N^2)log_2N)$ |
| FRFT filtering | $O((M + P + N)log_2N)$ |
| STFT reconstruction | $O((N^2 + N)log_2N)$ |
| sparse recovery | $O(N^2log_2N + KLN)$ |

*4.2. Experimental Results*

In this section, a Ku-band linear array radar system with 16 sensors and half-wavelength spacing is adopted. The pulse width of the transmitted signal is 10 µs and the bandwidth is 10 MHz. There is one target signal and two jamming signals. The target signal is from 0° with SNR of 12.5 dB. The first jamming signal is a deceptive jamming containing six false targets with INR of 30.5 dB, and the false targets occur after the real target. The second jamming signal is a deceptive jamming containing four false targets with INR of 22.3 dB, and the false targets occur before the real target. The experimental scenario is shown in Figure 14.

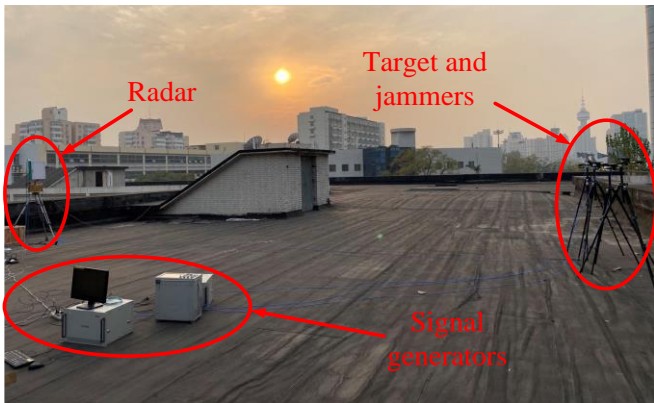

**Figure 14.** Experimental scenario.

Since both jammers transmitted full-pulse repeater jamming, the method in [4] is not suitable for jamming suppression. Then, the performance of the proposed method is evaluated and compared with the method in [11] and the method in [18]. The pulse compression results before and after proposed jamming cancellation method are shown in Figure 15. It can be seen from the figure that before jamming cancellation, the target signal is accompanied by multiple false targets; after jamming cancellation; however, the target peak is revealed, and the false targets are completely suppressed. It can be also seen that the comparison methods failed to completely cancel the jamming and the false targets are not fully suppressed. The SINR improvement based on 20 received pulses are shown in Figure 16 and the average SINR improvement is calculated and shown in Table 3. It is obvious that the proposed method can obtain better anti-jamming result compared with the other two methods and the average SINR improvement reaches more than 2.8 dB compared with other methods.

**Table 3.** Average SINR improvement of various methods.

| Method | SINR Improvement (dB) |
|---|---|
| Proposed | 15.86 |
| STFT reconstruction | 13.01 |
| Sparse recovery | 12.73 |

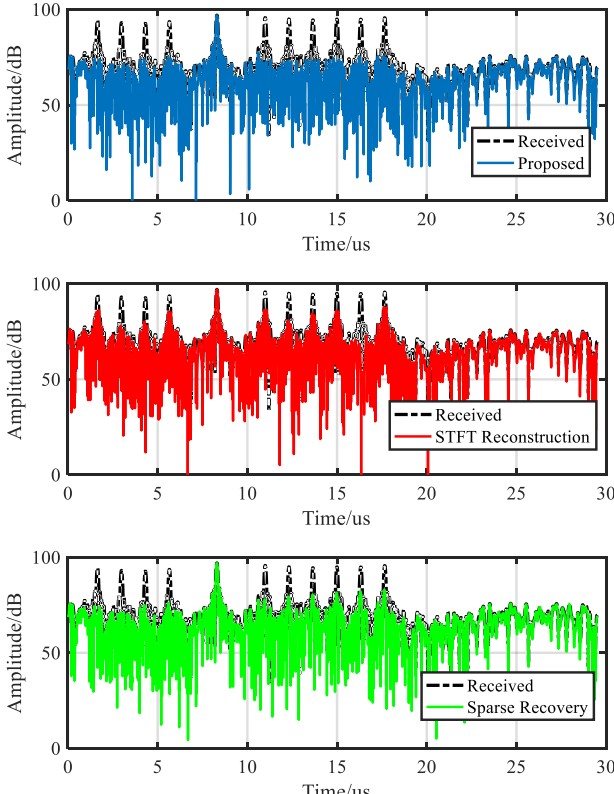

**Figure 15.** Pulse compression results of experimental data obtained by the proposed method, STFT reconstruction and Sparse recovery.

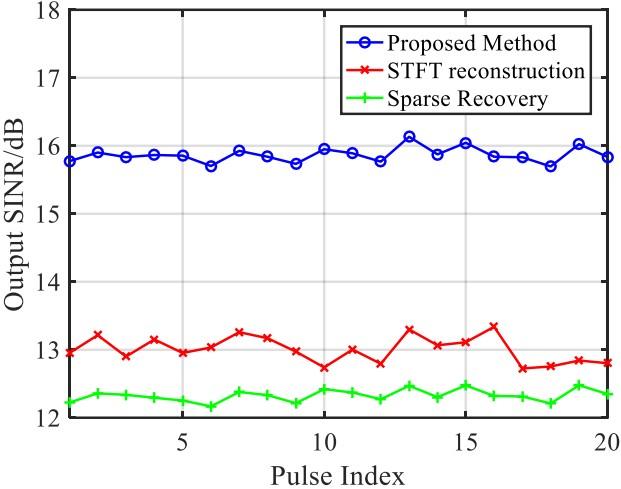

**Figure 16.** Comparison of SINR improvement of the proposed method with STFT reconstruction and Sparse Recovery using experimental data.

## 5. Conclusions

This paper presents a DRFM-based repeater jamming reconstruction and cancellation method with accurate edge detection. In the proposed method, basic jamming parameters are obtained by *STFRFT* spectrogram. On this basis, accurate jamming edges are estimated by combining DOB filtering result and TDDC curves. The normalized PC amplitude envelope of jamming can be reconstructed and canceled from the range profile of the received signal using the parameters estimated from the above steps. Numerical simulations and experiment are conducted to evaluate the algorithm's effectiveness for ISRJ and SMSP jamming suppression. The results verify that the proposed method can suppress these two types of jamming and the average SINR improvement reaches more than 2.8 dB compared with other methods.

**Author Contributions:** B.H. performed the theoretical study, conducted the experiments, processed the data and wrote the manuscript. X.Q. helped on the theoretical study, provided research suggestions and revised the manuscript. X.Y. provided the experiment equipment and revised the manuscript. W.L. and Z.Z. helped in performing the experiments and gave suggestions for the manuscript. All authors have read and agreed to the published version of the manuscript.

**Funding:** This research was supported in part by the National Natural Science Foundation of China (NSFC) under grants 61860206012 and 61901441.

**Data Availability Statement:** The data presented in this study are available on request from the corresponding author. The data are not publicly available due to confidentiality requirements.

**Conflicts of Interest:** The authors declare no conflict of interest.

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
