# Peer review of "DRFM-Based Repeater Jamming Reconstruction and Cancellation Method with Accurate Edge Detection"

_remotesensing, doi:10.3390/rs15071759_

Round 1

Reviewer 1 Report

Traditional jamming suppression methods do not attach importance to the accurate estimation of jamming edge, resulting in jamming residual and poor anti-jamming performance. To tackle this issue, this paper presents a DRFM-based repeater jamming reconstruction and cancellation method with accurate edge detection. In particular,numerical simulations and experiment are conducted to evaluate the algorithm’s effectiveness for ISRJ and SMSP jamming suppression. Compared with traditional methods, this method has certain performance advantages. I would like to propose some suggestions for revising the paper.

1. The pulse modulation rate of SMSP jamming should be n times of the original signal, considering the correctness of the FIG. 2.

2. It can be considered to omit formula (4) and give (5) directly.

3. The rect() function in formula (1) should be in standardized form, and such problems should be corrected in the whole article.

4. The signal amplitudes in formula (1) and formula (2) should be distinguished. Such problems should be corrected in the whole article.

Reviewer 2 Report

The authors are proposing a combination of methods, in particular the "difference of box" filters and "time domain deconvolution curves", as a novel jamming suppression method for DRFM-based repeater jamming, in order to accurately estimate the edges of DRFM jamming pulses. In this way, it is possible to achieve improved jamming reconstruction, obtaining better cancellation results. The proposed method is analysed and explained, while it is founded by numerical simulation and experiment results, proving that it offers a certain signal-interference-to-noise ratio (SINR) improvement.

The only remarks/suggestions are the following:

Line 12-13: «Traditional jamming suppression methods do not attach importance to the accurate estimation of jamming edge...» -> «Traditional jamming suppression methods do not give due importance to the accurate estimation of jamming edge...»

Line 19: different box -> difference of box

Line 361: signal -> signals

Concerning the term DRFM, this usually stands for «Digital Radio Frequency Memory», and not «Digital Radio Frequency Modulation». I propose that the authors should use the more standard nomenclature.

Finally, for the time unit, the «us» is used, instead of the correct «μs». If it is possible, please use the correct form with the greek letter «μ» (for micro).

Reviewer 3 Report

Jamming technics are widely used in electronic countermeasure war-field. For anti-jamming , authors have develop a processing structure carefully and make some valid efforts for depression DRFM-based repeater jamming .

1)in introduction section, a reference should be added about the intermittent sampling process.

2) the “PC domain” isn’t accurate for describing results of pulse compression processing.

3)sentences should be rewritten carefully according grammar and its’ syntactical structure. Eg, the verb word in “The second step is jamming signal reconstruction and cancellation”  should used as ‘are’

4)since Fig1 is same as Fig2, one figure is suggested for explanation different principles. 

5)In section 4.1, those names of comparison methods mentioned in ref4, ref11 and ref18 should be accordance with names in Fig10, Fig11, and shown directly.

6) authors should indicate where the calculations of SINR is proceeded and how to calculate.

7)  In fig14, those arrows seems indicating wrong direction.

1)considering computational complexity, what pros and cons does the method proposed have ? 2)Could authors give analysis of computational complexity ? 3)comparing with other algorithms, what innovative improvement has been made in paper ?
